



# Monitoring snow wetness evolution from satellite with Sentinel-1 multi-track composites

Gwendolyn Dasser[1,2,3], Valentin T. Bickel[4,5,6], Marius Rüetschi[5], Mylène Jacquemart[4,5], Mathias Bavay[1], Elisabeth D. Hafner[1,2,7], Alec van Herwijnen[1], and Andrea Manconi[1,2]

[1]WSL Institute for Snow and Avalanche Research, SLF
[2]Climate Change, Extremes and Natural Hazards in Alpine Regions Research Centre, CERC
[3]now at Department of Earth Sciences, Engineering Geology, ETH Zurich
[4]Laboratory of Hydraulics, Hydrology and Glaciology, ETH Zurich
[5]Swiss Federal Institute for Forest, Snow and Landscape Research, WSL
[6]now at Center for Space and Habitability, University of Bern
[7]EcoVision Lab, Photogrammetry and Remote Sensing, ETH Zurich

**Correspondence:** Gwendolyn Dasser (gwendolyn.dasser@erdw.ethz.ch)

**Abstract.** Information about snowpack wetness at high temporal and spatial resolutions is important for timely identification of pre-disposing conditions for avalanche release. However, such information is often available only for specific, instrumented locations. Space-borne techniques such as synthetic aperture radar (SAR) allow us acquiring information over large areas and in remote and challenging terrain. Here, we show how Sentinel-1 SAR multi-track composites can be used to monitor snow

wetness evolution over multiple seasons for a study site of $400\,\mathrm{km}^2$ around Davos in the eastern Swiss Alps. We validate the performance of our method using both in-situ measurements and modelled snowpack data. Moreover, we compared snow wetness maps and time series with wet avalanches records. We found correlations between SAR backscatter and modelled liquid water content between -0.25 and -0.59 for Spearman's rank coefficient and -0.25 and -0.64 for Pearson's correlation coefficient. By calculating the percentage of detected wet snow to dry/no snow per elevation, the season-elevation related

melting can be tracked. Moreover, we show that a rise of wet snow ratio above 40% coincides with an increase in wet snow avalanches releases in corresponding elevation bands. Our results suggest that wet snow products derived from Sentinel-1 SAR data may assist in identifying regions featuring a potential increase of wet snow avalanche activity. However, we could not find evidence of precursors of wet avalanche initiation with the accuracy required for operative monitoring applications.

## 1 Introduction

Wet snow avalanches can have a high destructive potential, however, they are still difficult to predict (Hendrick et al., 2023). In this respect, knowledge of the spatio-temporal distribution of snowpack wetness is of major relevance for wet snow avalanche hazard assessment, and more in general for accurate run-off modelling (Wendleder et al., 2018; Nagler et al., 2018; Dietz et al., 2012). Currently, avalanche warning is based on information from local observers, weather forecasts, and in-situ weather station measurements and spatial models (Hendrick et al., 2023). Temporally and spatially continuous high-resolution monitoring of

wet snow on large spatial scales and in complex terrain could be of great help to improve avalanche forecasting. Remotely



sensed data can help to overcome traditional challenges in poorly accessible terrains, thus providing cost-effective methods to efficiently monitor large and remote areas.

Space-borne synthetic aperture radar (SAR), such as implemented on the Sentinel-1 satellites, offers a promising approach to monitor snowpack evolution at regional scales. Specifically, changes in the backscattered radar signal can be used to provide

information on snow melt (Karbou et al., 2021; Truckenbrodt et al., 2019; Nagler et al., 2016), but the interaction between the snowpack and microwave frequencies is complex and not yet fully understood (compare Lievens et al., 2020). According to the current understanding, changes on SAR microwave signal can be measured only if the snowpack reaches several meters thickness (Dietz et al., 2012). Detailed sensitivity analyses between the interactions of SAR and snowpack characteristics is a focus of ongoing research (e.g., Lievens et al., 2019). On the other end, the presence of liquid water in the snowpack attenuates

the SAR backscattered amplitude (Tsai et al., 2019). This effect is useful to detect wet snow in SAR images by comparing signal backscatter amplitudes to a reference image without snow or with only dry snow (Nagler and Rott, 2000). When the signal decreases below a pre-defined threshold compared to the reference, wet snow is assumed to be present. By developing a representation style resolving the detected wet snow maps in altitude-time and altitude-orientation diagrams, Karbou et al. (2021) enabled to efficiently monitor the behaviour of wet snow lines across large spatial scale for avalanche prediction.

Publicly available products featuring wet snow coverage such as the one provided by European Environment Agency (2023), currently have several drawbacks, i.e.: (i) they are limited to a resolution of 60x60 m; (ii) they depend on multi-platform data fusion (optical and radar data) and therefore lack continuous, weather independent coverage; and (iii) they are prone to artifacts due to the SAR acquisition geometry. Regarding the latter, in mountainous terrain, the oblique, right-looking viewing geometry of the Sentinel-1 satellites leads to layover, shadowing and highly varying ground resolution. Shadowing effects

can be minimized by producing composites of radiometrically terrain-corrected images from ascending and descending orbits (Small et al., 2022). By combining data in this way, mountainous terrain is viewed from two different angles, thus allowing for compensation of missing data (shadow) in an individual orbit (Small et al., 2022). This approach of generating the so-called local resolution weighting (LRW) composites further minimizes outlier-effects and thereby reduces noise by averaging areas visible in both orbits. Furthermore, it allows to combine information gathered from different orbit thereby permitting to

increase temporal resolution to more than the orbit repeat interval.

In this study, we used LRW composites to mitigate the restrictions of traditional SAR images such as geometric effects. By optimizing the Sentinel-1 processing workflow, we improved temporal and spatial resolutions compared to current products such as Karbou et al. (2021). We applied the wet snow detection approach on a regional-scale $(400\,\mathrm{km}^2)$ in the area of Davos, Switzerland, and compared it to a data set of observed wet snow avalanches. We generated multi-track composites and over-

sampled the SAR data to match all data to 5x5 m ground sampling distance (GSD) over the study region. We validated the reliability of our results by cross-correlating the time series with data recorded from three weather stations as well as from the modelled snowpack information. With this study, we aim at providing information on wet snow avalanche preconditioning.





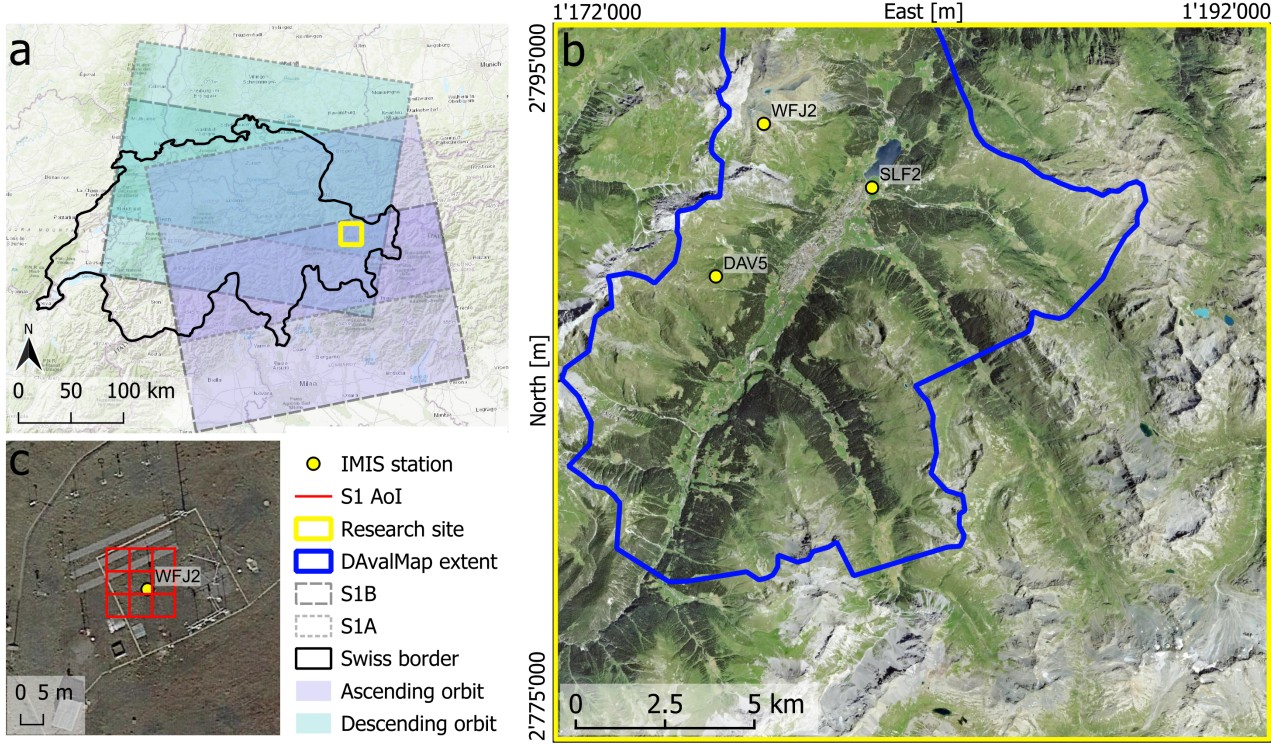

**Figure 1.** Overview of the area of research on three scales. (A) Outline of the Sentinel-1 footprints in relation also to the research site of interest (© Esri). (b) Area of study (yellow outline, displayed in reference system CH1903+/LV95, with EPSG: 2056, © Esri), extent and location of the avalanche library (DAvalMap, blue outline (Northwards cut to research area extent)), and the IMIS stations used (yellow dots). (c) Indication of the 3x3 pixel window used to calculate the Sentinel-1 backscatter median around the corresponding IMIS station (© Google Maps).

## 2 Data and methods

The study area is in the region of Davos, Switzerland (see Fig. 1), covering $400\,\mathrm{km}^2$ and elevation ranges between 1'542
and 3'225 m a.s.l. (according to Federal Office of Topography and Swisstopo, 2022). The site is characterized by steep, mountainous terrain and the presence of seasonal snow cover. Mean values of daily temperature in Davos (1'590 m a.s.l.) vary from $-5°$ to $13°C$ (MeteoBlue). The research area was selected due to its high alpine terrain as well as the availability of numerous reference data sets (station measurements and regular observations).

We used SAR data from the European Space Agency's Sentinel-1 satellites. The mission provides C-band data with a central
frequency of 5.405 GHz in the two polarizations VV and VH. Sentinel-1 has an orbital repeat cycle over central Europe every six to twelve days; however, depending on the location, revisit times can go down to 36 h, when exploiting a combination of all available tracks and orbits. Our analysis is based on the SAR images acquired on the relative orbits 66 (descending) and 15 (ascending; Fig. 1) in interferometric wide (IW) swath mode, where single-look-complex (SLC) data offer a pixel spacing



of ∼2.3 m in slant range and ∼14.1 m in azimuth (Bourbigot et al., 2016). The Sentinel-1 images over Davos are acquired
at 05:34 UTC on the descending track and at 17:15 UTC two days later on the ascending track, leading to a time difference
between the resulting composites of less than 60 h (in other parts of Switzerland a lag of 36 h can be achieved). The revisit
time in our data set is six days, but for current projects this would been reduced to 12 days due to the Sentinel-1B outage in
December 2021 (European Space Agency, 2022).

We applied radiometric terrain correction (gamma0, following Small et al., 2022) to descending and ascending acquisitions
and then combined the data into LRW composites. The latter products are calculated by considering a weighted function on at
least two different flight tracks (here one ascending and one descending). Weighting is applied according to the local spatial
resolution in the corresponding flight track (Small, 2012). We masked out areas that lie in radar shadow in both tracks, thereby
slightly diverging from the processing chain by Small et al. (2011) and Small et al. (2022). In cases where information is
available in only one track, i.e., lying in radar shadow in the other, only the available orbit was considered (as in Small, 2012;
Small et al., 2022). In total, we produced 206 LRW composites, 58 for the period August 2018 to July 2019, 58 for August
2019 to July 2020, and 59 for August 2020 to July 2021. Eight composites were discarded because one of the tracks was not
available (see Tab. A1). To minimize noise and artifacts, we calculated a temporal rolling median considering three samples,
thereby trading-off between a loss in detail and the influence of outliers.

Wet snow maps were generated by comparing an acquisition potentially featuring wet snow with a reference without snow
or with dry snow  (Nagler et al., 2016). Nagler et al. (2016) defined a threshold of 2 dB signal loss compared to a dry/no snow
image for Sentinel-1. In case of a stronger signal diminution, wet snow is assumed. No standard approach for the selection of
a reference image is yet established, leaving space for potential differences in results. As reference Nagler et al. (2016) used
a single image from summer, where the least amount of (melting) snow is assumed present in the scene. Rather than relying
on a single image, we calculate a median backscatter value per pixel over the entire available time series and use this value as
reference (indicated in black in Fig. 2). With this approach, we forego the challenge of finding a suitable reference image (see
Karbou et al., 2021), which becomes less accurate the longer the timespan between reference and acquisition becomes.

We additionally used data of elevation, forest and water bodies to mask out areas where wet snow detection with Sentinel-1
would be less reliable. A 5x5 m digital terrain model (DTM) provided by swisstopo was used (Federal Office of Topography
and Swisstopo, 2022). Aspect dependency analysis was performed by classifying the DTM into four aspect categories (N: 315-
45°, E: 45-135°, S: 135-225°, W: 225-315°). Information on the distribution of water bodies was taken from the high resolution
water and wetness (WAW) layer created by Copernicus (European Environment Agency, 2020). The data is provided at a 10x10
m grid scale and was acquired between 2012 and 2018 and published in 2020. The land classes were resampled to match the
Sentinel-1 pixel grid using nearest neighbour to assure the consistency of classes. The land cover classes of permanent water,
temporary water, permanently wet and temporarily wet were merged and used for masking. Information on forest cover in the
study area was sourced from the forest mask of the Swiss national forest inventory (Waser et al., 2015). Spatial filtering was
performed using a spatial median moving window filter with a kernel size of 3x3.





## 2.1 Reference data

Two different validation data sets were used. First, we analyzed the sensitivity of SAR backscatter to snowpack characteristics to identify the features influencing Sentinel-1 signal by computing the correlation with respect to a variety of both measured 100 and modelled parameters. Second, we compared the wet snow maps from Sentinel-1 to recorded wet snow avalanches.

### 2.1.1 Modelled Snowpack characteristics from SNOWPACK

The SNOWPACK snow cover model has been developed to support operational avalanche warning service based on the inter-cantonal measurement and information system (IMIS) automatic weather station data (Lehning et al., 1999). It models snow as a three-phase porous medium made of ice, water and air with an arbitrary number of layers in the underlying soil and in the 105 snow (glacier ice when present is treated as snow layer with the physical properties of ice). It features a very detailed description of snow properties including weak layer characterization (Stössel et al., 2010), phase changes, water transport in snow with a simplified model (Hirashima et al., 2010) or with full Richards Equations (Wever et al., 2014) and water vapor transport in snow (Jafari et al., 2022). SNOWPACK is used by several countries for their operational avalanche warning services (Morin et al., 2020) but also in research. SNOWPACK is provided alongside its meteorological preprocessor MeteoIO (Bavay and 110 Egger, 2013) and Graphical User Interface Inishell (Bavay et al., 2022) under an open source licence (LGPLv3).

In this study, we used the SNOWPACK version from December 2022 (git version: 98a23cd) in an operational setup. This means that the simulations did not model soil, relying on a Dirichlet lower boundary condition (upper soil temperature). Moreover, the upper boundary condition (energy exchange between the snow or soil and the atmosphere) relied on either a measured snow, a soil surface temperature (Dirichlet when the surface temperature is less than -1 °C) or a parameterized 115 incoming long wave radiation (Neumann) that evaluates the atmospheric cloudiness based on the measured short wave radiation (Carmona et al., 2014). These reanalyses additionally used a horizon computed from a 25 m resolution digital elevation model in order to distinguish shading by the terrain from cloudy sky, when parameterizing the incoming long wave radiation. The SNOWPACK output includes both measured data at the nearest station as well as modelled parameters. The IMIS stations SLF2 (1563 m a.s.l.), at DAV5 (2315 m a.s.l.) and WFJ2 (2536 m a.s.l.) are within the boundaries of the research site (Fig. 120 1), the IMIS measurements are hence covering different elevations within the region of interest in a flat field simulation. Timespan covered by available SNOWPACK data is September 2017 to August 2020. Due to a sensor failure, data from SLF2 are not available for season 2018-19 and hence the corresponding SNOWPACK data was not used in this analysis due to lower reliability.

Our SNOWPACK runs provided outputs every 3 hours and we used the time closest to Sentinel-1 acquisition on the cor-125 responding day: namely 06:00 to match descending SAR image acquisition and 18:00 to match the ascending acquisition (UTC+1). We averaged the data from these two SNOWPACK timestamps, refraining from calculating a composite according to the applied weighting in the SAR images, as the IMIS stations are generally situated in flat terrain and so the weighting of the two orbits is similar. SNOWPACK data was run between January 2018 and August 2020.



Finally, and in order to test the factors influencing the SAR backscatter time series, we performed a cross-correlation analysis between diverse variables in SNOWPACK against at the corresponding station location over time (see appendices A3, A4 and A5). We considered Pearsons correlation coefficient including RMSE values, as well as Spearmans rank coefficient. To minimize correlation coefficient bias due to the effect of over-represented numerical values (e.g. zero is used by SNOWPACK when void), entries occurring more than 25% of the time were ignored in the correlation analysis. Significance level for both tests were set to p-values below 0.05.

After detection, we classified wet snow within different elevation classes, and calculated the ratio of pixels identified as wet snow in comparison to no/dry snow images following the plot design by Karbou et al. (2021). Elevation classes have been set to 100 m and the range defined between 1500 and 3000 m a.s.l., resulting in 15 different elevation steps. Temporal classes were aligned according to the ascending Sentinel-1 acquisitions. Areas with a slope angle below 28° have been masked out to focus on avalanche release areas (Bühler et al., 2018). We calculated the ratio by dividing the amount of wet snow pixels by the total amount present pixels available at each elevation band.

### 2.1.2 Wet snow avalanche catalogue

As a further ground reference, we considered data from the Davos Avalanche Mapping Project (DAvalMap, detailed description in Hafner et al. (2021)). This data set has been set up by the swiss avalanche warning service and is designed to be an avalanche inventory covering about 180 km². Systematic mapping has been performed within the illustrated perimeter in Fig. 1 (Hafner et al., 2021) and is thus smaller than our study area. A part of DAvalMap in the North has been excluded to match the research area, which has been fitted to be within the national boundaries for data consistency. Mappings outside the DAvalMap are only sporadic. Spontaneous avalanches considered small according to their type (50 m for slab avalanche and glide-snow avalanche, 100 m for loose snow avalanches in either dimension) are not recorded, thereby setting a minimum size of avalanches within the data set. The data includes information on the release date, the snow conditions (wet/dry snow avalanche) and the elevation zone at release.

For this study, we select only the avalanches that are classified as wet snow avalanches and are situated within the research site (see Fig. 1) and time range (January 2018 to August 2021). The recorded avalanches from DAvalMap were matched to wet snow ratio plots according to the elevation of the release zone and the date of release (Karbou et al., 2021). Temporal matching was performed by pooling the observed avalanches into one elevation-time bin by combining the avalanches occurring up to three days before ascending image acquisition, the ones occurring at the acquisition date itself and until to two days after the ascending image acquisition (for illustration see appendix Fig. A2). This allowed to cover a continuous time frame by means of a six day interval. This temporal matching also includes avalanches after image acquisition. However, this is a robust solution to approximate the wetness state of the snowpack and its corresponding proneness to wet snow avalanches for it minimized temporal differences between avalanches to acquisitions. In season 2019-20 the size restriction of the recorded avalanches was handled less strict also including smaller loose snow avalanches which would have been excluded in the other years. The used data from DAvalMap contains 77, 723 and 261 wet snow avalanches for the seasons 2018-19, 2019-20 and 2020-21, respectively.





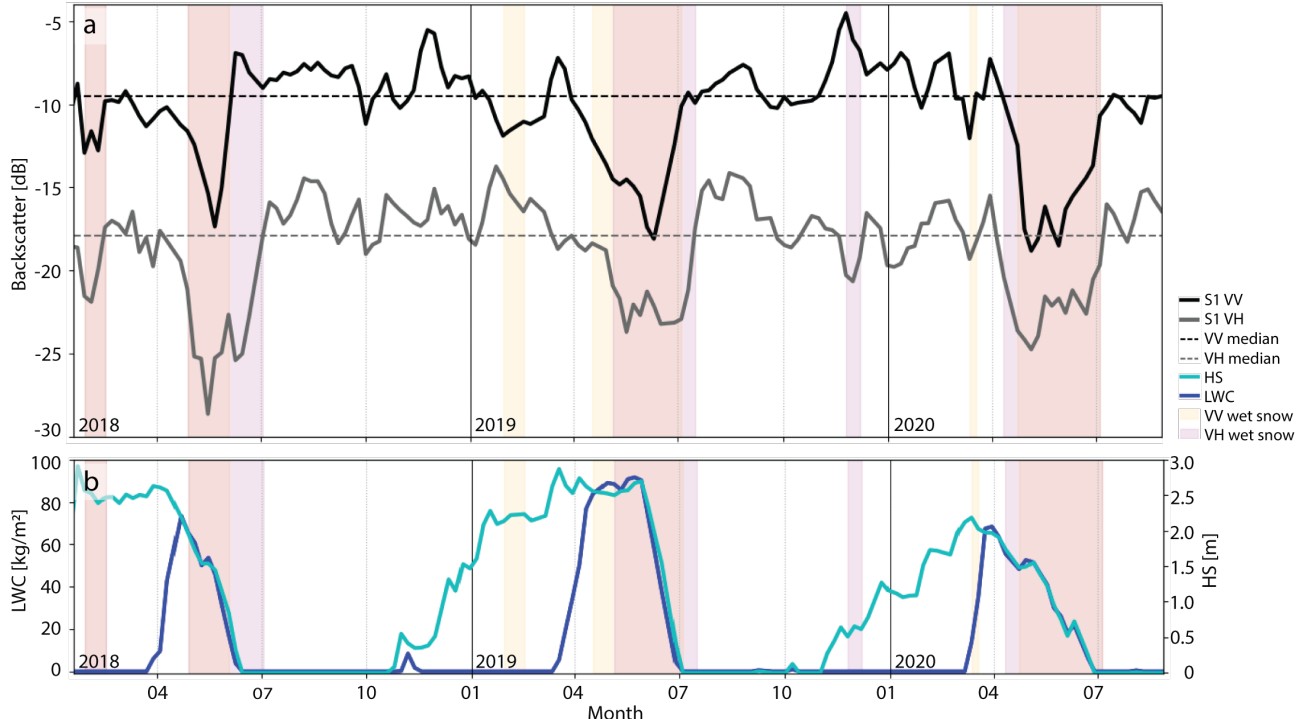

**Figure 2.** Time series of wet-snow evolution from January 2018 until August 2020 at Weissfluhjoch (WFJ2). The upper panel (a) shows the time series extracted from the pixel within which the IMIS station is situated in both co- and cross-polarization states using a three sample rolling mean over the time series. The horizontal line indicates the time series median and the shading background displays the times during which the backscatter falls lower than 2 dB below the calculated median in the corresponding polarization. The lower panel (b) shows the time series of measured liquid water content (LWC) and measured snow height (HS) at the station as extracted from SNOWPACK data.

## 3 Results

### 3.1 Polarization dependent Sentinel-1 backscatter time series analysis

The time series for both the VV and VH polarization over two and a half years at the IMIS station of WFJ2 are shown in Fig. 2a, with the corresponding median value indicated with dashed lines. Backscatter falling 2 dB below the median (see Fig. 2a) is categorized as presence of wet snow (indicated in shading in Fig. 2). Generally, the cross-polarized data shows a lower backscatter value (e.g. see Fig. 2a), its median at WFJ being 8.35 dB below the median of the co-polarized data (see Tab. 1). The median backscatter of the DAV5 station is found several dB lower than the other stations (see Tab. 1).

At WFJ, the backscatter in both polarizations (VV & VH) fell clearly (> 3 dB) below the set threshold for wet snow in late spring/early summer in each season, thereby coinciding with snow melt in high alpine altitudes. Sporadic signal losses reaching below the threshold can be found, where SNOWPACK data does not suggest presence of snow wetness, however only one of such events (February 2018) is detected in both polarizations. In other spurious events, such as February 2019



**Table 1.** Median of rolling mean of Sentinel-1 backscatter per polarizations at the three different stations.

| Polarization | SLF2 | DAV5 | WFJ2 |
|---|---|---|---|
| VV | -10.13 | -14.19 | -9.54 |
| VH | -17.71 | -20.25 | -17.88 |

or December 2019, results diverge depending on the polarization. The combination of VV and VH polarization (indicated by overlaid shadings in Fig. 2) shortens the timespan of detected wet snow, while minimizing the detection during off-season periods. While similar results were found in the station SLF2, the time series of the DAV5 station does not show a clear trend but rather extreme short lived peaks and hence unstable signal. This also results in lower backscatter median as shown in Tab. 1.

Increasing Liquid Water Content (LWC) percentage coincides with a decrease in snow height (see Fig. 2b). In each melting season an increase in LWC can be found as well as sporadic peaks matching early snow melting events e.g. in November 2018. With a time shift of a couple of days the backscatter signal diminishes soon after the sudden increase in LWC and the decrease in snow height. This behaviour can be found across all three melting seasons (see Fig. 2).

Indication for negative correlation between S1 backscatter and SNOWPACKS modelled liquid water contents could be found across all stations, polarizations and kernel sizes (see appendices A3, A4 and A5). These correlations ranged between -0.25 to -0.59 (spearmans rank) and -0.25 to -0.64 (Pearsons) with an RMSE between 2.05 and 4.35 (see appendix A3). At WFJ2 station, S1 indicated a positive correlation with SNOWPACK's potential albedo, however this could not be found in the other stations locations.

## 3.2 Spatial distribution and elevation dependent melting

Time-elevation dependent melting seasonality was detected across the time series independent of polarization and slope aspect (Fig. 3): In season 2018-19 the highest wet snow ratios were measured (VV polarization, mid March, classified into wet snow ratio of 80-90 %). Increasing ratio values (above 0.2) were detected earlier in the season of 2019-20 between September to May in elevations ranging from 1800 to 2400 m a.s.l. The maximum ratios that were reached throughout this season amounted to 60-70 %. In 2020-21 (see Fig. 3) the wet snow ratio rose again, achieving up to 70-80 % between end of April to end of March. Wet snow ratio mainly diverged between polarization states in late summer and autumn periods, where VV polarization indicates maximum wet snow ratio values of up to 2 %, whereas VH polarization indicates a continuous time-elevation dependent increase in wet snow ratio with up to 40 % wet snow pixels per elevation class. VH indicates higher ratio values than VV polarized data in lower elevation classes (1500-1900 m a.s.l.).

Masking of landcover classes such as forest and water bodies had little influence on the resulting time-elevation plots. Large water bodies such as lakes are mainly situated in the lowest classes and in flatlands, thereby already getting masked out by focusing on hillsides via the minimal slope threshold (hence not extra shown in App. A1). On the other end, the exclusion



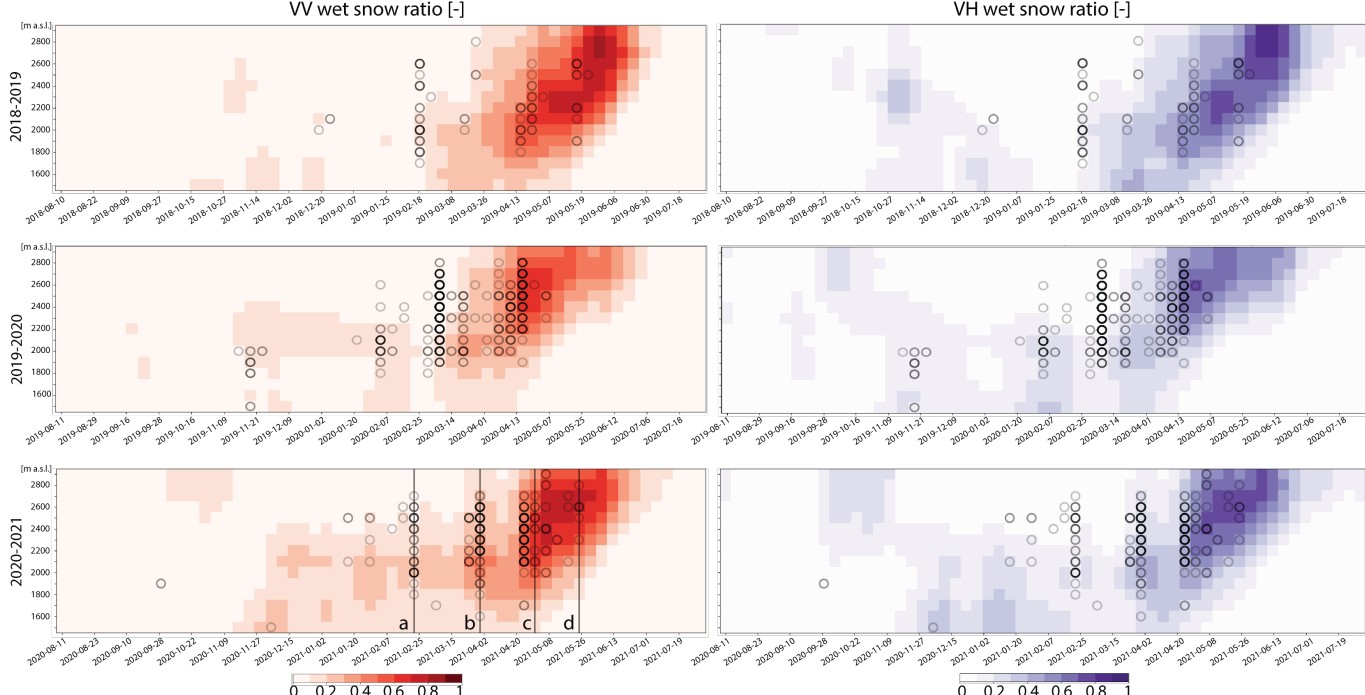

**Figure 3.** Time-elevation plots of wet snow ratio evolution for VV and VH polarization across three seasons (2018-19, 2019-20 and 2020-21) featuring the area of Davos as indicated in Fig. 1, resolved at 100 m elevation bands. The wet snow ratio is the percentage of pixel per elevation band that was detected as wet snow using S1 data. Circles indicate a top-down view of a histogram featuring the recorded wet snow avalanches per time step according to the DAvalMap data set (Hafner et al., 2021). The hue of the circles corresponds to the avalanche count, black corresponds to 4 or more avalanches. Forest areas were masked out and a moving median filter was applied to minimize the ffect of noise. Letters a-d correspond to the timeframes of the wet snow maps displayed in Fig. 5. Plot design was inspired by Karbou et al. (2021) and Karbou et al. (2022)

of forest results in measurable changes in elevation-based wet snow mapping (see App. A1.A to B) as they also occur in non-flat areas. Forests are known to influence radar backscatter and cover a larger and more diverse landscape than water bodies. However, also their influence is mainly focused on a specific elevation range below 2000 m a.s.l. (see App. A1.A to B) Applying a moving median filter allows to smooth the time-elevation plots (see App. A1.A to C). The achieved noise reduction has less impact in high alpine areas. This is likely because less signal disturbance is present in the first case. Elevation classes in lower elevation regions result in classification differences between with and without noise reduction of two classes (from 0-10 % to 30-40 % wet snow ratio values).

The time-elevation plots also clearly shows differences in aspects (see Fig. 4). South and East facing slopes showed an earlier increase in wet snow ratio to values above 40 % in the season than West and especially North facing slopes (compare Fig. 4.B and C to A and D). Highest ratios reaching 90-100 % wet snow ratio values are mainly located in North facing slopes, where melting occurs later in the season and over a shorter time period. The same behaviour across time-elevation can be found in





VH polarized data as well as across all seasons. The aspect-dependency is more relevant if a minimal slope steepness to e.g. 45° (high avalanche danger according to Bühler et al. (2018)) is defined.

Wet snow ratios do not show relevant changes before the first surge of wet avalanches recorded in DavalMap (see Figs. 3a, 4
and 5). Instead, a second increase of avalanche occurrence observed in 2018-19 and 2020-21 ranging across various elevation classes (Fig. 3b), is coinciding with a wet snow ratio values of 30 to 40 % in altitudes from 2000-2200 m a.s.l. After this second event a further increase in wet snow ratio characterizes the melting season (Fig. 3). Events of multiple avalanche occurrences recorded in DavalMap coincide with wet snow ratios above 40 % (Fig. 3).

## 4   Discussion

In this study, we analyzed SAR-based wet snow maps by implementing a new processing chain and validating it with in-situ measurements and modelled information. By relying on LRW composites, Sentinel-1 data has been exploited to combine different orbits as well as different tracks. Thus, the information on snow wetness in our study is provided with higher frequency compared with methods used at present (European Environment Agency, 2023).

The available Sentinel-1 based snow cover products SAR Wet Snow or Wet/Dry Snow by Copernicus offer a spatial grid
sampling distance of 60x60 m (European Environment Agency, 2023) and hence may not suffice to adequately resolve the variability of alpine topography. Additionally, as the Wet/Dry Snow layer fuses Sentinel-1 and -2, it is restricted to i) cloud-free Sentinel-2 acquisitions and to ii) dates were both Sentinel-1 and -2 were acquired. This results in a restricted temporal availability of the product (European Environment Agency, 2023). The discrepancy between the available products and our results is partly linked to the use of the higher resolved SLCs instead of the ground range detected Sentinel-1 European Environment
Agency (2023); Karbou et al. (2021). Moreover, by using LRW composites, the influence of typical SAR geometry effects, such as shadowing and layover, could be minimized (see Fig. 5). Having fewer areas which lack data and a better temporal resolution is important, for example for avalanche forecasting where accurate timing of capturing the start of the wetting of the snowpack, a key factor to estimate the hazard of wet snow avalanche releases.

Validation of spatially continuous products providing information, where there was previously hardly any, is challenging.
Consequently, we relied on comparing data with in-situ measurements, where available, to minimize uncertainties in the reference data. The SAR Wet Snow and Wet/Dry Snow layer were validated by a pixel-wise comparison to other remote sensing products such as snow cover maps from Landsat (European Environment Agency, 2023; Karbou et al., 2022). Since the acquisition technique is still based on remote sensing approaches the biases contained might also be similar, hence we believe our comparison to an independent product, though not covering the same area, is less prone to bias. Also, we compare the results
to snowpack characteristics instead of snow extent, thereby more reliably approximate what is actually influencing the signal.

The SAR backscatter is influenced by many factors, therefore to identify wet snow a threshold needs to be set in comparison to a dry/no snow reference. This threshold is usually set between 2 and 3 dB signal loss relative to a reference image or annual median (this work) and is based on histogram analysis (e.g., Nagler et al., 2016). Adaptive thresholds based on the underlying land cover class (Liu et al., 2022) have recently been suggested. However, we have used the established and widely



applied threshold of 2 dB for Sentinel-1 data (Nagler et al., 2021, 2016, results indicated in Fig. 2). Even though an adaptive backscatter analysis sounds promising, there are no widely available and regularly updated land cover products with suitable spatial resolution available for our case and we try to minimize the input data.

As previously mentioned, no standard approach for the generation of a reference image has been established. The Copernicus SAR Wet Snow layer is based on a reference image calculated as the median from a stack of summer acquisitions (assumed no snow; European Environment Agency, 2023). Other studies calculate the mean over dry or no snow images (e.g., Karbou et al., 2021) or use a single reference image featuring either dry or no snow conditions (e.g., Mendes et al., 2022). Increasing temporal baselines between the image of interest and the reference image, also result in increase of errors due to changes over time. In the application of wet snow avalanche now- and forecasting time efficiency and accuracy is of major importance. By choosing an all season median, we have wet snow contamination in the reference; however the results of our study indicate applicability of the method (see Fig. 2) and follow the process applied in Lievens et al. (2019). The advantage of reference image generation in that way are low processing costs, low amount of input, high automation level, data and easy update-ability.

Validation has been performed using SNOWPACK data at three IMIS stations SLF2, DAV5, WFJ2 (appendixes A3, A4 and A5). This allows us to include both modelled and measured variables, thereby also allowing to check the validity of the gained results. The stations are all situated on mountain slopes but in rather flat areas. Hence, they are unlikely to accurately represent the conditions in highly variable topography limiting the possibility to validate LRW products. Additional limitations may come from the immediate surroundings of some of the stations having an impact on the radar backscatter, such as the infrastructures around the WFJ2 station.

We have tested the cross-correlation of values recorded on a single pixel to diverse variables contained in SNOWPACK at IMIS station locations. However, single pixel analyses are prone to radar noise (e.g., speckle) and uncertainties, we have therefore performed the same analysis on a 3x3 window (indicated in Fig. 1) and at three different locations. Across all tests, the liquid water content was among the strongest correlation values (see appendixes A3, A4 and A5). This coincides with the ongoing understanding of the measured signal loss being related to the amount of liquid water within a snowpack (e.g., Lund et al., 2022; Nagler et al., 2021). With increasing water presence, radar attenuation increases and measured signal decreases. Already with a LWC of 5 % in a snowpack, SAR penetration depth is found to be minimized to one wavelength and hence SAR backscatter is already very low (Lund et al., 2022; Nagler et al., 2016).

The correlation values show similar trends between the different polarization modes (see appendixes A3, A4 and A5). The strongest divergence between the wet snow ratio extracted from VV and VH polarization can be found in late summer early autumn, when the VH polarization indicates higher ratio values than VV. This might be related to a remaining influence of seasonal activity of vegetation (e.g. by non-masked shrub forest). The time series shows similar results for both polarization modes. A combination of both polarization's as done by e.g. Karbou et al. (2021) could result in fewer misclassification, however, the approach will also result in a loss of detail. A separate analysis of the polarization state also solves the issue of using a thresholding approach developed by Nagler et al. (2016) on a dB scale on a ratio value as done by (Karbou et al., 2021). Future research might examine the potential of a hotspot approach based on separate mapping of wet snow per polarization state for wet snow monitoring.



We observed a time lag between the decrease in snow height, the associated increase in LWC, and the detection of wet snow from Sentinel-1 (see Fig. 2). We believe this gap may be attributed to the combination of four aspects: i) the temporal gap between variables, as we make use of Sentinel-1 data acquired only every 6-12 days, ii) the potential filtering out of the onset of wetting in a snowpack as noise due to temporal or spatial filters applied, iii) the limited sensitivity of the sensor up to a certain minimum water content threshold and iv) the dependence of the abundance of the LWC on the acquisition time of, in

our case, descending images as the snowpack might refreeze during a clear cold night. If and how these factors contribute to the miss of the first avalanche surge needs to be investigated further in the future.

Our results show an earlier start of the melting season in lower elevation zones with increasing melting later in the season in higher elevation zones (see Fig. 3). This agrees with the station data, where at the higher situated stations winter lasts longer and they have a more pronounced and later melting season. Lower elevations typically have a shorter and warmer winter season.

This is leading to a shorter period of signal attenuation during the melting phase than in stations with more snow coverage in higher elevation zones.

The DAvalMap library contains recorded avalanches within the area of the perimeter indicated in Fig. 1 (Hafner et al., 2021). Multiple aspects must be considered when interpreting the wet snow ratio per elevation to this data set: i) The monitored perimeter is smaller than the research site, so avalanches outside of the parameter are not systematically monitored and mapped.

This results in elevation bins above approx. 2800 m a.s.l. to contain little to no recorded avalanche activity. ii) The avalanches used in the study were all wet snow avalanches, so the snowpack in the starting zone of the avalanches was assumed wet. Since this classification of snow wetness in the starting zone is challenging from a distance, some avalanches might have been wrongly included. We assume that this influenced the miss of the first wet snow avalanche surge in our analysis, where the avalanches had wet snow deposits even though they started out as dry snow avalanches and we only considered start zone

coordinates. Later when the snowpack is wet from the valley to the mountaintops, this potential classification error becomes negligible. iii) The information is based on in-situ observations dependent on good visibility, so the assigned release date may include uncertainties up to several days. However, with our Sentinel-1 timeframe of six days the influence will be minimal. Despite those challenges, DAvalMap provides a unique opportunity to compare observed activity with snow wetting from Sentinel-1 data.

The first surge of wet snow avalanches could not be matched to a relevant increase in the wet snow ratio across all seasons (see Fig. 3). This might be due to i) uncertainties in the DAvaMap library such as dry/wet classification errors (see above), ii) the use of a median potentially biased by wet snow as reference value, iii) an insufficient capability of temporally resolving the avalanche release dynamics based on the used SAR data.

We encourage future research to make use of not only multi-platform but also multi-frequency data such as a combination

of X- and L-band with the upcoming NISAR mission (Oveisgharan et al., 2024). This would allow to capture different stages of the melting due to differences in sensitivity to water presence with varying wavelengths, thereby enabling to better estimate the LWC and not only the presence of liquid water.



## 5 Conclusions

We used Sentinel-1 SAR multi-track composites to derive snow wetness at different elevations and slope aspects in alpine
areas. By exploiting LRW composites created on SLC data, we were able to more accurately resolve spatial variability of the snow wetness in alpine terrain. We validated the performance of our approach by cross-correlating the data to in-situ measured, modelled snowpack variables and a wet snow avalanche inventory, finding relevant correlation between Sentinel-1 backscatter and modelled liquid water content, as well as good agreement with release of wet snow avalanches coincided with the wetting of the corresponding elevations, with the exception of the first wet snow avalanche surge. Our results indicate
potential of Sentinel-1 based monitoring of wet snow avalanche preconditioning factors provided the availability of data with higher spatiotemporal resolution than provided by the Sentinel-1 orbit repeat interval

Follow-up research in the area of Davos should make use of other available Sentinel-1 tracks, namely 117 and 168 to improve temporal resolution to the available revisit time instead of being restricted to the orbit repeat interval. This could help understand the underlying mechanisms better, even though the multi-orbit data is only available for selected regions. Despite
this, our results suggests that the wet snow ratio per elevation bin calculated from SAR data may in the future help increase the accuracy of forecasting wet snow avalanches by nowcasting wet snow distribution also over large, challenging and remote areas.



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

*Data availability.* SAR data can be downloaded from the Copernicus Open Access Hub or from the Alaska Space Facility. SAR processing has been performed using the gamma software. The DAvalMap library and SNOWPACK data for WFJ station can be requested from SLF.





**Figure 4.** Time-elevation plots of wet snow evolution for VV polarization for the exemplary seasons of 2020-2021. Displayed are the time series resolved into the different aspects: A: North, B: South, C: East and D: West. The plot encompasses data featuring the area of Davos as indicated in Fig. 1, resolved at 100 m elevation bands. The wet snow ratio is the percentage of pixel per elevation band that was detected as wet snow using S1 data. Circles indicate a top view of a histogram featuring the recorded wet snow avalanches per time according to the DAvalMap data set (Hafner et al., 2021). Letters a-d correspond to the timeframes of the wet snow maps displayed in Fig. 5. Plot design was inspired by Karbou et al. (2021) and Karbou et al. (2022)



**Figure 5.** Spatial distribution of the wet snow ratio and the recorded avalanches in DAvalMap (Hafner et al., 2021) are displayed for four exemplary, different time intervals (a-d) as indicated in Fig. 3 and Fig. 4. The acquisition times of the SAR images are indicated in the title per subplot on the left, while the DAvalMap indicates the time frame of summarized wet snow avalanches.



**Appendix A**

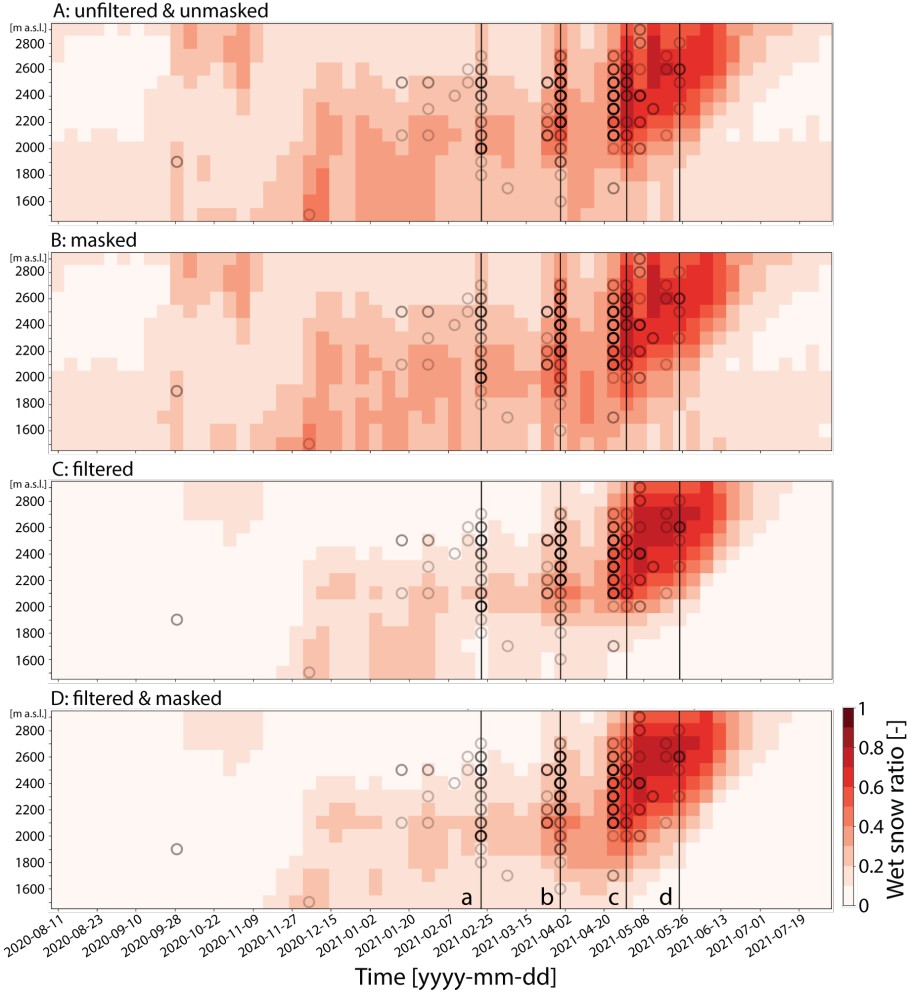

**Figure A1.** Time-elevation plots of wet snow evolution following Karbou et al. (2021) and Karbou et al. (2022) for VV polarization for the exemplary seasons of 2020-2021. It shows the effect of filtering and masking: A) no filtering or masking applied, B: forest mask applied, C: spatial median filter with a window size of 3x3 applied, D: forest mask and spatial filtering combined (same as in 3). The plot encompasses data featuring the area of Davos as indicated in Fig. 1, resolved to 100 m spatial elevation classes. Circles indicate a top view of a histogram featuring the recorded wet snow avalanches per time according to the DAvalMap data set (Hafner et al., 2021). Letters a-d correspond to the timeframes of the wet snow maps displayed in Fig. 5.



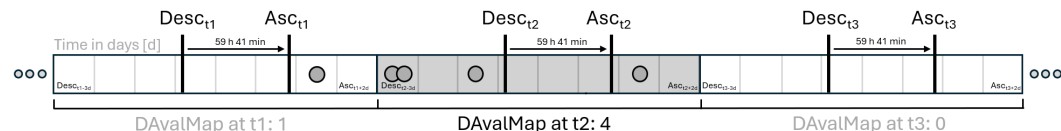

**Figure A2.** Exemplary illustration of performed temporal binning featuring the SAR acquisition times in descending (Desc) and ascending (Asc) track to corresponding local resolution composites (shading) in combination with the summed wet snow avalanches. Binning has been optimized to include all recorded wet snow avalanches (from DAvalMap catalogue, indicated in circles) while minimizing the temporal offset between an acquisition towards the avalanche to best match the wet snow condition.





**Table A1.** Dates of acquisition of ascending orbit information of the discarded composites.

| | |
|---|---|
| 02.03.2019 | 16.09.2019 |
| 20.01.2021 | 01.02.2021 |
| 07.02.2021 | 03.03.2021 |
| 15.03.2021 | 27.03.2021 |





**Table A2.** Abbreviation table for the variables from SNOWPACK, which are also used in the following tables containing the cross-correlation variables.

| | |
|---|---|
| Measured relative humidity | RH |
| Measured air temperature | TA |
| Measured surface temperature | TSS_meas |
| Modelled surface temperature | TSS_mod |
| | |
| Modelled hoar size | hoar_size |
| Measured snow height | HS_meas |
| Modelled snow height | HS_mod |
| Measured wind direction | DW |
| Measured wind velocity | VW |
| Measured wind drift velocity | VW_drift |
| Modelled 24h wind | wind_trans24 |
| | |
| Modelled rain rate | MS_Rain |
| Modelled solid precipitation | MS_Snow |
| Modelled snow water equivalent (SWE) | SWE |
| Modelled total liquid water content (LWC) | MS_Water |
| Modelled sublimation | MS_Sublimation |
| Modelled evaporation | MS_Evap |
| Modelled erosion mass loss | MS_Wind |
| Modelled surface mass flux | MS_SM_Flux |
| Modelled snow runoff (virtual lysimeter) | MS_SN_Runoff |
| | |
| Potential albedo | pAlbedo |





**Table A3.** Cross correlation analysis table of the analysed SNOWPACK variables compared to the median VV Sentinel-1 backscatter time series at the WFJ2 station (location see Fig. 1). The table includes the results per polarization state for 3x3 window size (15x15 m) and 1x1 pixel size (5x5 m). Included are the Spearmans (Spearm.) and Pearsons (Pears.) correlation coefficient and the RMSE calculated between the Pearsons and the actuall data. * indicate values that came with a p-value of below 0.05. N is the amount of available data per variable used for the correlation. Abbreviations can be found in A2

| | N | 3x3 VH | | | 3x3 VV | | | 1x1 VH | | | 1x1 VV | | |
|---|---|---|---|---|---|---|---|---|---|---|---|---|---|
| | | Spearm. | Pears. | RMSE | Spearm. | Pears. | RMSE | Spearm. | Pears. | RMSE | Spearm. | Pears. | RMSE |
| RH | 149 | -0.1 | -0.05 | 3.38 | 0.09 | 0.05 | 3.2 | -0.1 | -0.07 | 3.73 | 0.06 | 0.05 | 3.56 |
| TA | 149 | 0.06 | -0.03 | 3.38 | -0.01 | -0.05 | 3.2 | 0.02 | -0.05 | 3.73 | -0.02 | -0.06 | 3.56 |
| TSS_meas | 149 | 0.12 | 0.07 | 3.38 | 0 | -0.01 | 3.21 | 0.07 | 0.04 | 3.73 | -0.02 | -0.03 | 3.57 |
| TSS_mod | 149 | 0.11 | 0.07 | 3.38 | -0.02 | 0 | 3.21 | 0.06 | 0.04 | 3.73 | -0.04 | -0.03 | 3.57 |
| hoar_size | 41 | 0.24 | 0.22 | 2.96 | 0.17 | 0.06 | 2.75 | 0.25 | 0.21 | 3.33 | 0.16 | 0.04 | 3.38 |
| HS_meas | 102 | 0.05 | 0.07 | 3.48 | -0.24* | -0.21* | 3.54 | 0.01 | 0.03 | 3.92 | -0.26* | -0.23* | 3.94 |
| HS_mod | 102 | 0.1 | 0.13 | 3.45 | -0.21* | -0.17 | 3.56 | 0.06 | 0.09 | 3.9 | -0.24* | -0.2* | 3.97 |
| DW | 149 | -0.01 | -0.04 | 3.38 | -0.11 | -0.13 | 3.18 | -0.01 | -0.05 | 3.73 | -0.12 | -0.14 | 3.53 |
| VW | 149 | -0.19* | -0.13 | 3.35 | -0.17* | -0.15 | 3.17 | -0.17* | -0.12 | 3.71 | -0.16 | -0.14 | 3.53 |
| VW_drift | 149 | -0.24* | -0.13 | 3.36 | -0.13 | -0.1 | 3.19 | -0.21* | -0.12 | 3.71 | -0.11 | -0.09 | 3.55 |
| wind_trans24 | 70 | -0.07 | -0.05 | 2.94 | 0.21 | 0.24 | 3.25 | -0.1 | -0.08 | 3.28 | 0.16 | 0.23* | 3.74 |
| MS_Rain | 33 | -0.03 | 0.12 | 3.6 | -0.04 | 0.07 | 4.1 | -0.03 | 0.12 | 3.81 | -0.02 | 0.07 | 4.49 |
| MS_Snow | 28 | -0.11 | 0.05 | 2.82 | 0.17 | 0.04 | 4.01 | -0.06 | 0.06 | 3.02 | 0.11 | 0.08 | 4.44 |
| SWE | 102 | -0.07 | -0.05 | 3.48 | -0.38* | -0.36* | 3.37 | -0.1 | -0.08 | 3.9 | -0.39* | -0.38* | 3.75 |
| MS_Water | 52 | -0.3* | -0.27* | 3.42 | -0.45* | -0.43* | 3.46 | -0.3* | -0.25 | 3.91 | -0.47* | -0.42* | 3.72 |
| MS_Sublimation | 36 | -0.02 | -0.05 | 3.01 | 0.06 | -0.01 | 2.95 | 0.01 | -0.02 | 3.31 | 0.12 | 0 | 3.62 |
| MS_Evap | 28 | -0.32 | -0.35 | 2.92 | -0.09 | -0.12 | 4.01 | -0.35 | -0.38* | 3.31 | 0 | -0.05 | 4.36 |
| MS_Wind | 7 | 0.04 | -0.01 | 2.34 | 0 | 0.17 | 2.58 | 0.18 | 0.02 | 2.42 | -0.07 | 0.18 | 2.9 |
| MS_SM_Flux | 45 | -0.27 | -0.17 | 3.59 | -0.12 | -0.02 | 4.13 | -0.25 | -0.17 | 4.06 | -0.07 | 0 | 4.4 |
| MS_SN_Runoff | 36 | -0.33* | -0.37* | 3.05 | -0.31 | -0.25 | 3.81 | -0.29 | -0.34* | 3.72 | -0.25 | -0.23 | 4.11 |
| pAlbedo | 103 | 0.48 | 0.31* | 3.3 | 0.32* | 0.09 | 3.58 | 0.46* | 0.3* | 3.72 | 0.29* | 0.1 | 4.01 |

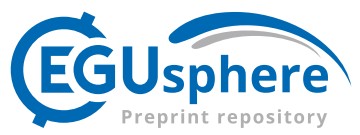

**Table A4.** Cross correlation analysis table of the analysed SNOWPACK variables compared to the median VV Sentinel-1 backscatter time series at the DAV5 station (location see Fig. 1). The table includes the results per polarization state for 3x3 window size (15x15 m) and 1x1 pixel size (5x5 m). Included are the Spearmans (Spearm.) and Pearsons (Pears.) correlation coefficient and the RMSE calculated between the Pearsons and the actuall data. * indicate values that came with a p-value of below 0.05. N is the amount of available data per variable used for the correlation. Abbreviations can be found in A2.

| | N | VH 3x3 | | | VV 3x3 | | | VH 1x1 | | | VV 1x1 | | |
|---|---|---|---|---|---|---|---|---|---|---|---|---|---|
| | | Spearm. | Pears. | RMSE | Spearm. | Pears. | RMSE | Spearm. | Pears. | RMSE | Spearm. | Pears. | RMSE |
| RH | 149 | -0.16* | -0.14 | 3.21 | 0.02 | 0.07 | 2.36 | -0.19* | -0.15 | 3.85 | -0.02 | 0.06 | 2.78 |
| TA | 149 | 0.01 | 0.03 | 3.24 | 0.06 | 0.05 | 2.36 | 0.05 | 0.06 | 3.89 | 0.08 | 0.06 | 2.78 |
| TSS_meas | 149 | -0.03 | -0.01 | 3.25 | 0.08 | 0.08 | 2.35 | -0.01 | 0 | 3.9 | 0.07 | 0.07 | 2.78 |
| TSS_mod | 149 | -0.17 | -0.19 | 3.43 | -0.19 | -0.24* | 2.22 | -0.15 | -0.16 | 4.17 | -0.11 | -0.13 | 2.63 |
| hoar_size | 42 | -0.24 | -0.3 | 2.61 | 0.01 | 0 | 1.86 | -0.23 | -0.23 | 3.38 | 0.02 | 0.08 | 2.53 |
| HS_meas | 92 | 0.27* | 0.23* | 3.51 | -0.13 | -0.15 | 2.34 | 0.18 | 0.14 | 4.19 | -0.2 | -0.21* | 2.57 |
| HS_mod | 92 | 0.28* | 0.23* | 3.5 | -0.07 | -0.11 | 2.36 | 0.18 | 0.13 | 4.19 | -0.17 | -0.2* | 2.58 |
| DW | 149 | 0.05 | 0.01 | 3.25 | -0.05 | -0.01 | 2.36 | 0 | -0.03 | 3.9 | -0.04 | 0.04 | 2.78 |
| VW | 149 | 0.03 | -0.02 | 3.25 | -0.18* | -0.2* | 2.32 | 0.04 | 0.02 | 3.9 | -0.2* | -0.2 | 2.72 |
| VW_drift | 149 | 0.02 | -0.02 | 3.25 | -0.18* | -0.2* | 2.32 | 0.03 | 0.01 | 3.9 | -0.21* | -0.2* | 2.72 |
| wind_trans24 | 18 | 0.36 | 0.44 | 3.92 | 0.21 | 0.25 | 2.17 | 0.24 | 0.38 | 4.82 | 0.17 | 0.17 | 2.75 |
| MS_Rain | 34 | -0.17 | -0.1 | 2.82 | 0.28 | 0.01 | 2.7 | -0.04 | 0.08 | 3.49 | 0.31 | 0.04 | 3.41 |
| MS_Snow | 17 | 0.28 | 0.17 | 4.14 | -0.19 | -0.19 | 2.71 | 0.36 | 0.3 | 4.82 | -0.21 | -0.26 | 3.02 |
| SWE | 92 | 0.18 | 0.13 | 3.57 | -0.18 | -0.22* | 2.31 | 0.12 | 0.07 | 4.22 | -0.24* | -0.27* | 2.54 |
| MS_Water | 38 | -0.44* | -0.41* | 3.77 | -0.59* | -0.64* | 2.05 | -0.4* | -0.38* | 4.35 | -0.42* | -0.52* | 2.48 |
| MS_Sublimation | 27 | -0.02 | -0.11 | 2.91 | 0.03 | 0.1 | 1.64 | -0.04 | -0.09 | 3.75 | 0.11 | 0.1 | 2.23 |
| MS_Evap | 17 | 0.15 | 0.28 | 3.76 | -0.36 | -0.24 | 2.48 | 0.11 | 0.26 | 4.31 | -0.31 | -0.07 | 2.74 |
| MS_Wind | 7 | -0.5 | -0.18 | 4.37 | 0 | 0.22 | 2.61 | -0.36 | -0.19 | 4.94 | 0 | 0.34 | 3.09 |
| MS_SM_Flux | 43 | 0.02 | -0.02 | 3.46 | 0.19 | 0.08 | 2.92 | 0.06 | 0.05 | 4.08 | 0.15 | 0.01 | 3.17 |
| MS_SN_Runoff | 24 | 0.17 | 0.23 | 3.75 | -0.21 | -0.06 | 2.69 | 0.27 | 0.31 | 4.26 | -0.17 | -0.06 | 2.85 |
| pAlbedo | 92 | 0.12 | 0.02 | 3.6 | 0.23* | 0.15 | 2.34 | 0.02 | -0.06 | 4.22 | 0.1 | 0.06 | 2.63 |





**Table A5.** Cross correlation analysis table of the analysed SNOWPACK variables compared to the median VV Sentinel-1 backscatter time series at the SLF2 station (location see Fig. 1). The table includes the results per polarization state for 3x3 window size (15x15 m) and 1x1 pixel size (5x5 m). Included are the Spearmans (Spearm.) and Pearsons (Pears.) correlation coefficient and the RMSE calculated between the Pearsons and the actuall data. * indicate values that came with a p-value of below 0.05. N shows the amount of available data per variable used for the correlation.Abbreviations can be found in A2

| | N | VH 3x3 | | | VV 3x3 | | | VH 1x1 | | | VV 1x1 | | |
|---|---|---|---|---|---|---|---|---|---|---|---|---|---|
| | | Spearm. | Pears. | RMSE | Spearm. | Pears. | RMSE | Spearm. | Pears. | RMSE | Spearm. | Pears. | RMSE |
| RH | 111 | -0.07 | 0 | 2.24 | -0.03 | -0.01 | 2.28 | -0.04 | 0.01 | 2.53 | 0 | 0.01 | 2.57 |
| TA | 111 | 0.25* | 0.23* | 2.19 | -0.13 | -0.09 | 2.28 | 0.22* | 0.22* | 2.47 | -0.13 | -0.08 | 2.57 |
| TSS_meas | 111 | 0.26* | 0.22* | 2.19 | -0.15 | -0.11 | 2.27 | 0.23* | 0.21* | 2.48 | -0.15 | -0.1 | 2.56 |
| TSS_mod | 111 | 0.14 | 0.09 | 2.23 | -0.1 | -0.07 | 2.28 | 0.14 | 0.1 | 2.52 | -0.1 | -0.06 | 2.57 |
| hoar_size | 19 | 0.66* | 0.5* | 2.41 | -0.16 | -0.21 | 2.56 | 0.67* | 0.5* | 2.78 | -0.16 | -0.23 | 2.82 |
| HS_meas | 45 | -0.08 | -0.06 | 2.55 | 0.28 | 0.24 | 2.9 | -0.14 | -0.13 | 2.9 | 0.22 | 0.2 | 3.2 |
| HS_mod | 50 | -0.13 | -0.12 | 2.47 | 0.27 | 0.21 | 2.81 | -0.18 | -0.16 | 2.81 | 0.24 | 0.18 | 3.09 |
| DW | 111 | 0.01 | 0.02 | 2.24 | 0.05 | 0.07 | 2.28 | 0.02 | 0.05 | 2.53 | 0.06 | 0.08 | 2.57 |
| VW | 111 | 0.08 | 0.05 | 2.24 | 0.03 | 0.04 | 2.28 | 0.06 | 0.01 | 2.53 | -0.01 | 0.02 | 2.57 |
| VW_drift | 111 | 0.08 | 0.05 | 2.24 | 0.03 | 0.04 | 2.28 | 0.07 | 0.01 | 2.53 | -0.01 | 0.02 | 2.57 |
| wind_trans24 | 20 | 0.04 | -0.08 | 2.74 | 0.16 | -0.03 | 3 | 0.07 | -0.09 | 3.37 | 0.25 | -0.03 | 3.53 |
| MS_Rain | 44 | 0.14 | 0.16 | 2.05 | -0.02 | 0 | 2.11 | 0.14 | 0.14 | 2.42 | -0.04 | -0.02 | 2.26 |
| MS_Snow | 5 | 0.9* | 0.87* | 1.02 | 0.2 | -0.36 | 1.98 | 0.9* | 0.82 | 1.77 | 0.2 | -0.11 | 2.86 |
| SWE | 30 | -0.22 | -0.24 | 2.42 | 0.16 | 0.08 | 2.86 | -0.25 | -0.25 | 2.75 | 0.14 | 0.05 | 3.14 |
| MS_Water | 38 | -0.36* | -0.4* | 2.18 | -0.26 | -0.33* | 2.77 | -0.37* | -0.36* | 2.52 | -0.25 | -0.33* | 2.98 |
| MS_Sublimation | 14 | -0.15 | -0.17 | 1.77 | 0.16 | 0.16 | 2.11 | -0.07 | -0.15 | 2.19 | 0.14 | 0.21 | 2.23 |
| MS_Evap | 18 | 0.22 | 0.16 | 2.34 | -0.17 | -0.14 | 2.79 | 0.22 | 0.18 | 2.56 | -0.23 | -0.13 | 2.98 |
| MS_Wind | 1 | NaN | NaN | NaN | NaN | NaN | NaN | NaN | NaN | NaN | NaN | NaN | NaN |
| MS_SM_Flux | 44 | 0.22 | 0.1 | 2.13 | 0.18 | 0.13 | 1.94 | 0.23 | 0.08 | 2.34 | 0.14 | 0.11 | 2.29 |
| MS_SN_Runoff | 18 | 0.1 | 0.07 | 2.14 | 0.06 | 0.16 | 2.34 | 0.02 | 0.05 | 2.58 | 0.14 | 0.22 | 2.61 |
| pAlbedo | 54 | 0.14 | 0.07 | 2.51 | 0.23 | 0.1 | 2.79 | 0.13 | 0.03 | 2.84 | 0.28* | 0.12 | 3.05 |



*Author contributions.* Conceptualization: AM, MJ; Data Curation: AM, GD; VB; Formal Analysis, Investigation, Software, Visualization: GD, VB; Methodology: GD, VB, AM, MJ; Project Administration: AM, MJ; Validation: GD, VB, MB, EH; Writing – Original Draft Preparation: GD; Writing – Review & Editing: GD, VB, AM, MJ, MR, MB, EH, AvH; Funding Acquisition: MJ, AvH; Supervision: AM

*Competing interests.* The contact author declares that none of the authors has competing interests.

*Acknowledgements.* This work has been financed by the WSL research program for Climate Change Impacts on Mass Movement (CCAMM,
https://ccamm.slf.ch/). The authors thank David Small for discussions on SAR geometry issues including local resolution weighing and multitrack composites, as well as Martin Hendrick for discussions on SNOWPACK variables and Yves Bühler for his contribution on funding acquisition.