# Peer review of "Monitoring snow wetness evolution from satellite with Sentinel-1 multi-track composites"

_EGUsphere, 2024_

## Author Comment (AC1)

**Response to Reviewer 1**

Dear Reviewer 1,

We sincerely appreciate the time and effort you have invested in reviewing our manuscript. Below, we address your comments (colour-coding: blue review comment, black answer statement).

**Major Comments**

1. **Clarification of Methods**
   a. **In Situ/Model Data Description**

I found the description of the in-situ (AWS data) and SNOWPACK model to be rather confusing. Are the IMIS stations assimilated into SNOWPACK? Does the SNOWPACK timeseries at station locations reflect the station data or if not, how has it been modified? Are the timeseries in Figure 2 observed or modelled data? Please see minor comments below for additional detail.

We acknowledge that the description of the in-situ AWS data and SNOWPACK model could be clearer. All data used in our study were extracted from SNOWPACK rather than directly from station measurements. SNOWPACK is forced by the meteorological measurements performed by the AWS and assimilates the snow height as measured at the station (ultrasonic snow height sensor). As a detailed energy balance model with an arbitrary number of layers, it is able to simulate the stratigraphy of the snow at the AWS (as necessary for avalanche warning applications) such as weak layers or the wetting of the snow (Wever et al., 2016)). We will revise the manuscript accordingly to ensure this distinction is clearly communicated. We suggest adding the following at line 104 in the manuscript to make it clearer:
"The snow cover model SNOWPACK simulates the detailed stratigraphy of the snowpack based on meteorological input data from an automatic weather stations. Specifically, the model uses local meteorological measurements, such as air temperature, snow depth or snow surface temperature, to describes the snow microstructure, density, temperature and liquid water content of the layers in the snowpack"

   - **Wet Snow Ratio**

The wet snow ratio (ln 139) features prominently in the results but is fairly "hidden" in the methods. I suggest presenting this as a formal equation to help highlight it and then reference it elsewhere in the manuscript.

We agree that the wet snow ratio plays a central role in our study and should be highlighted more prominently in the Methods section. We will introduce a formal equation for its calculation ensuring a clear reference point throughout the manuscript.

2. **Impact of Sentinel-1 Overpass Timing (Morning and Evening) on Merged Product**

The manuscript describes that a key advance of this workflow is the use of LRW composites to overlook viewing geometries. However, snowmelt and LWC varies temporally over multiple temporal scales (diurnal to seasonal). What is the impact of including morning and evening overpasses in the LRW composites? What if a pixel is frozen during the morning acquisition but melting in the evening acquisition?

We recognize the importance of this point and appreciate the reviewer's concerns. However, we argue that if the impact of this issue were substantial, it would manifest as systematic differences between east- and west-facing slopes in steep terrain. This is because east-facing slopes receive more weight from the ascending (morning) acquisitions, while west-facing slopes are more influenced by descending (evening) acquisitions. No such systematic bias has been observed in our

data (and was also not observed in prior studies performed by the authors, see Dasser (2021)), supporting our assumption that the impact is minimal.

Additionally, to rigorously assess this effect related to the overpass timing, we would require independent reference data that are both accurate and spatially representative of steep terrain. We focused on cross-correlation between station data and the LRW composites, demonstrating that the backscattered signal is indeed driven by snow wetness. Given that our study's primary goal is to assess the feasibility of using Sentinel-1 for wet snow avalanche preconditioning rather than validating the LRW product itself, we consider a more detailed analysis of Sentinel-1 acquisition timing beyond the scope of this manuscript. We have already mentioned the issue in the discussion section (lines 283-286)

**3. More Detailed Comparison to Previous Work**

The discussion starts out describing the advance of this work over prior methods (lines 220-224) but this comparison was not explicitly made in the manuscript. I suggest the addition of a rigorous comparison to existing workflows be included, as that would demonstrate this advance more clearly. This in regards to both temporal (lines 220-224) and spatial (lines 224-226) scales.

We acknowledge that the discussion in lines 220-224 could better compare our approach to previous studies. To address this, we will incorporate all four Sentinel-1 tracks into our dataset, further emphasizing the improvements in temporal coverage. Additionally, we will include the SAR Wet Snow data corresponding to Figure 5, enabling a clear visual comparison that also highlights spatial enhancements.

**4. Data Availability**

I strongly encourage the authors to archive the datasets in a publicly available repository rather than by request from an organization. At present, the manuscript is not compliant with TC data policy: https://www.the-cryosphere.net/policies/data_policy.html, "The best way to provide access to data is by depositing them (as well as related metadata) in FAIR-aligned reliable public data repositories, assigning digital object identifiers, and properly citing data sets as individual contributions."

We recognize the importance of open and FAIR-aligned data sharing, in accordance with *The Cryosphere*'s data policy. To ensure compliance, we will make our code available via GitHub and data accessible through EnviDat (link) upon publication.

**Minor Comments**

2 – snow wetness has implications beyond avalanche release, as detailed later. A broader justification for this work would be appropriate in the abstract.

While we acknowledge that snow wetness has broader implications beyond avalanche release, we prefer to maintain a focused discussion on its relevance for wet snow avalanches. We will clarify this focus in the title and manuscript, as also suggested by Reviewer 2.

3- replace "allow us" to "facilitate"

Line 3          Done, thanks.

4- replace "show how" with "utilize"

Line 4          Done, thanks.

4 – delete "can be used"

Line 4          Done, thanks.

Line 5          Done, thanks.

5 – I suggest using evaluate rather than validate here and elsewhere (ln 50, for example)

Line 5          Done, thanks.

Line 10         Explicitly define the wet snow ratio as the amount of pixels per elevation band that were classified as wet snow over the amount of all pixels present in corresponding elevation band.

14 – simplify "operative monitoring application" to operational monitoring

Line 14         Done, thanks.

15 – include specifics on why wet snow avalanches are still difficult to predict

Line 15         This difficulty can be attributed to the limited availability of in-situ measurements of liquid water content, as well as the incomplete understanding of how liquid water influences the mechanical properties of the snowpack. (Hendrick et al., 2023). We will explain this better in the manuscript.

23-34 – I found this paragraph to be somewhat confusing, as background on the use of SAR for SWE retrievals and snowmelt detection is presented.  I suggest revising to clarify which application is being discussed.

Lines 23-34     We will improve clarity in the discussion of SAR applications and focus on snowmelt detection.

Line 26         We will revise ambiguous phrases such as "compare Lievens et al. 2020" for better clarity

Line 30-34      Thanks for pointing out these additional references. We will evaluate if they are relevant to our study and include them accordingly.

48 – I suggest including details on the Karbou et al. 2021 products rather than just the citation here.

Line 48         We will describe details on the Karbou et al. 2021 products

Figure 1 caption - revise: …of research at three spatial scales.

Done, thanks.

Line 54-55      We will change formatting of elevation values (from 3'225 to 3225 m a.s.l. for consistency)

55 – provide more details regarding field site slope. I recognize that this is a large area, but a range and median value would provide a useful characterization.

Line 55        We will add range & median value for field site slope for a better feeling of the area.

55 – provide additional details on forest cover (tree types, canopy heights, % of study area, etc)

Line 55        We will provide additional details on forest cover (tree types, canopy heights, % of study area, etc)

58 – Does this range represent daily values over the entire year? Averaged over how many years?

Line 58        We will clarify origin of provided temperature values

67 – clarify: this would be reduced

Line 67        Rephrased to: "The revisit time in our dataset is six days; however, for current projects, it has extended to 12 days due to the Sentinel-1B outage in December 2021 (European Space Agency, 2022)."

80-81 – provide more details from the Nagler et al study on why this 2dB threshold was selected

Line 80-81    Explain why the 2 dB threshold was selected based on Nagler et al. (2000)

                The selection of this threshold featured in the discussion section lines 241-247. As referenced in the manuscript we based our selection on Nagler et al., 2016, which is when the authors have published an adjusted threshold of 2 dB specifically for Sentinel-1 applications based on their histogram analysis (rather than from Nagler and Rott, 2000 where the threshold was set to 3 dB for ERS data). Since this is current state of the art and references are provided, we believe that further explanation in the manuscript is not necessary.

83 – revise sentence structure to not start with: As reference Nagler…

Line 83        Done, thanks.

94 – what percent forest cover was used for this mask? How sensitive were the results to this selection?

Line 94        We will clarify the forest cover threshold used for masking

133 – Please clarify and provide an example for what is meant by "entries occurring more than 25% of the time were ignored"?

Line 133      We will provide a clearer explanation of the 25% threshold for data exclusion

                We can rephrase this to make the statement clearer. 0 and NaN values (depending on the parameter, NaN if 0 was a sensible value such as in temperature values and 0 when it did not naturally occur in the parameter) were overrepresented across space and time and drastically influenced the output statistics (e.g. 0 in snowheight, which was most of the time throughout the hydrologic year). Such values were discharged from the assessment.

136 – please describe this plot design in more detail for readers not familiar with Karbou et al. 2021

Line 136      We will add detailed description of this plot design rather than referring to Karbou et al. 2021

138 – Why were the temporal classes defined by the ascending rather than descending (or average) Sentinel-1 acquisitions?

Line 138    We have selected ascending for illustrating purposes, however we have adjusted for this in the matching process as illustrated in fig. A2.

138 – Was the sensitivity to masking by 28 degrees assessed?  Given radar viewing geometries on steep slopes, could better results be achieved by assessing wet snow presence on slopes <28 degrees even though those don't coincide with avalanche starting zones?

Line 138    Our study focuses on using Sentinel-1 data for wet snow avalanche preconditioning. Slopes below 28° were excluded to maintain practical relevance, as they are less critical for avalanche release. Including them would suggest higher accuracy than what can be achieved in the more challenging alpine terrain relevant for our field of application, where spaceborne SAR faces greater limitations.

139 – I suggest referring to it as the number of pixels rather than amount of pixels.

Line 139    Use "number of pixels" instead of "amount of pixels"

153 – could add a reference to the wet snow ratio equation (see previous comments)

Line 153    add a reference to the wet snow ratio equation which will be newly implemented

159 – check formatting throughout regarding en-dashes

Line 159    Check formatting regarding en-dashes to follow the manuscript according to The Cryosphere guidelines: change 2019-20 to 2019–20 throughout.

Fig 2- what is the temporal resolution of the LWC plotted in Figure 2?  3 hrs or has this been smoothed to daily resolution? If smoothed, does the original time series include diurnal variability?

Fig 2    As mentioned in the text we have averaged the SNOWPACK data to best match the S1 LRW. We here calculated the mean between the data available closest to the acquisitions in asc and desc geometry – with an operational setup of SNOWPACK featuring 3h output interval. With this, we attempted to calculate the closest data that actually influenced the radar signal displayed in the upper part of the plot. We can attempt to rephrase in the text and mention correspondingly in the figure caption.

Fig 2 – in 2020, the LWC (blue) increases when HS increases in the spring?  HS increases indicate snowfall with <0 C temperatures, so why would LWC increases occur simultaneously? This can also be seen in 05-2018.

The SNOWPACK model gets its mass input from two sources: either by assimilating the snow height or from rain gauges. When assimilating the snow height, by default an air temperature threshold of 1.2°C is used to discriminate between rain and snow (it is also possible to instead rely on a linear interpolation between -2°C and +2°C to produce mixed precipitation). When using data from rain gauges and if the rain gauges are not heated, their data will be discarded except when the conditions are such that liquid precipitation can occur (relative humidity high enough, difference between the air temperature and the snow surface temperature below a given threshold). This allows SNOWPACK to get accurate estimates of mass input (thanks to the snow height assimilation) while also being able to handle rain on snow events (by using rain gauge data in such conditions; Bavay et al., 2024).

Moreover, SNOWPACK uses an internal timestep of 15 minutes, although in its operational mode (as used for the operational avalanche warning) it only write its outputs once every 3 hours, thus sometimes blurring the picture of its handling of the various processes.

In the Spring, it can happen that the air temperature is hovering over the rain/snow temperature threshold and thus generating either rain or snow at each 15 minutes timesteps that will be seen as mixed precipitation when accumulated over 3 hours outputs. Moreover, the station's unheated rain gauges might also provide true mixed precipitation to SNOWPACK. If SNOWPACK is forced with mixed precipitation, both the snow height and the liquid water content might increase simultaneously.

We will Improve clarity in the Figure 2 caption, ensuring it distinguishes between model output and measured data (line 2)

176-77 – given the reference to the other stations, a similar time series to what is shown in Figure 2 should be included as supplementary figures.

Line 176-77     We will provide similar figures in the supplementary figures for the other two stations.

179 – LWC doesn't need to be capitalized when being defined in acronym.

Line 179        Done, thanks.

181 – "a time shift of a couple of days" is vague yet this offset is important to understand. I suggest quantifying this in a more rigorous manner.

Line 181        Quantify the time shift in snowmelt detection instead of using "a couple of days"

185 – capitalize Spearman's

Line 185        Done, thanks.

190 – insert comma after 2018-19,  also replace dash with en dash

Line 190        Done, thanks.

193 – clarify: end of April to end of March. Is this meant to be end of May?

Line 193        Good catch! Corrected to "end of April to end of May"

200 – what is meant by hence not extra?

Line 200        This note was meant to indicate that, since water bodies are already excluded by the minimum slope mask of 28°, we did not include an additional subplot in the appendix to specifically assess their influence. However, this detail is not essential and can be removed.

257-260 – If the SNOWPACK model is assimilating the station data (which is what I understand is happening), what is the value in only using the model output at the station locations over the measured station data?   Further, why not compare the model output over the full domain to the Sentinel-1 melt products through time?

Line 257-260      The stations only assimilate HS, and relies on meteorological forcings to simulate all snow properties. The stations don't measured LWC or SWE, we will make this clearer throughout the manuscript, so without SNOWPACK it would not have been possible to show the correlation with LWC. For HS, since the model output data at the station location is identical to the measured station data, there is neither a gain nor a loss in using it. We do not compare the model output to the full domain over time because the accuracy of the model decreases as the distance from the station increases. Consequently, comparing a remotely sensed dataset to a modelled dataset over the full domain would not provide meaningful insights, as it would be unclear which dataset is closer to the actual conditions.

265 – was the sensitivity to the selected 3x3 window evaluated?  What if a 5x5 window was utilized? This might provide some insight to the previous statement regarding whether "stations having an impact on the radar backscatter."

Line 265      We have tested different window sizes (e.g., 5×5) and found minimal effects on radar backscatter. However, as no method is entirely free from spatial dependencies, we discuss this potential limitation while opting for the higher-resolution 3×3 solution. We will clarify this reasoning in the manuscript.

280 – see previous comment re: time lag

Line 280      Done, thanks.

294 - should this read perimeter rather than parameter?

Line 294      Yes! We have corrected to perimeter, thanks.

298 – revise sentence for improved clarity "this influenced the miss of the first…."

Line 298      Done, thanks.

326 – delete also

Line 326      Done, thanks.

**Additional Comments**

1. **Discussion on Avalanche Dynamics and Snowmelt Percolation**

I suggest the addition of a Discussion paragraph on avalanche dynamics related to meltwater percolation. Sentinel-1 is sensitive to surface to near-surface melt while wet snow avalanches initiate due to failure of a buried interface or at the bottom of the snowpack. The discussion would benefit from some insights regarding these differences.

We appreciate the reviewer's request to clarify Sentinel-1's sensitivity to different snowpack layers. We note that Sentinel-1 is not exclusively sensitive to surface or near-surface meltwater (Strozzi & Mätzler, 1998).  Given the attenuation properties of the radar signal, a buried melt layer or basal meltwater can also act as a dominant specular reflector, resulting in significant backscatter loss. We will include a brief discussion of these effects in the manuscript. Also, while we agree with the reviewer on wet-snow avalanches release is cased due to water percolation deeper in the snowpack, the water has to be produced at the snow surface (Mitterer & Schweizer, 2013)

2. **Rain-on-Snow Events and Avalanche Frequency**

Another common trigger of wet snow avalanches is rain on snow. While it would be beyond the scope of the manuscript to add a significant analysis in this regard, a simple analysis comparing a re-analysis precipitation product (e.g., ERA5-Land) with the avalanche frequency might identify which events are melt related and which were precipitation related, and improve the comparisons presented in this manuscript.

We agree that rain-on-snow is a critical factor in wet snow avalanche formation and acknowledge that a comparison with a reanalysis precipitation dataset (e.g., ERA5-Land) could provide valuable insights. However, the relationship between rain-on-snow events and avalanche occurrence is complex and not as straightforward as suggested (Nander et al.; Würzer et al., 2017). Incorporating a new dataset introduces further, dataset specific uncertainties.

Instead, we will consider analysing the number of recorded wet snow avalanches in the reference catalogue in relation to the temporal distribution of all liquid precipitation averaged over the study area. Such an approach would provide an initial assessment of the potential influence of rain-on-snow events while minimizing additional data usage.

**3. Data Availability**

Data Availability: See major comment. Also provide a reference for the Gamma software program.

We will add the reference to the software version used and specifically to the Sentinel-1 implementation into the software (Wegmüller et al., 2016). Thanks.

**References**

Bavay, M., Wever, N., Fierz, C., & Lehning, M. (2024). Looking back at the last 15 years of operational avalanche warning with the snowpack model in Switzerland. In K. Gisnås, P. Gauer, H. Dahle, M. Eckerstorfer, A. Mannberg, & K. Müller (Eds.), *Proceedings of the international Snow Science Workshop 2024* (pp. 82-87). Norwegian Geotechnical Institute.

Dasser, Gwendolyn. (2021). Master Thesis: "Analysis of Multi-Track Backscatter Time Series for Cryospheric Applications and their Feasibility for Snow Depth Classification." Department of Geography, University of Zurich. Access via: https://lean-gate.geo.uzh.ch/typo3conf/ext/qfq/Classes/Api/download.php/mastersThesis/734

Mitterer, C. & Schweizer, J. (2013) "Analysis of the snow-atmosphere energy balance during wet-snow instabilities and implications for avalanche prediction". *The Cryosphere*, 7, 205–216, doi:10.5194/tc-7-205-2013.

Strozzi, T. & Mätzler, T. (1998). "Backscattering Measurements of Alpine Snowcovers at 5.3 and 35 GHz." *IEEE TRANSACTIONS ON GEOSCIENCE AND REMOTE SENSING*, vol. 36, no. 3, may 1998.

Wegmüller, U., Werner, C., Strozzi, T., Wiesmann, A., Frey, O. & Santoro, M., "Sentinel-1 Support in the GAMMA Software", *Procedia Computer Science*, Volume 100, 2016, Pages 1305-1312, ISSN 1877-0509, https://doi.org/10.1016/j.procs.2016.09.246.

Wever, N., Vera Valero, C., & Fierz, C. (2016)."Assessing wet snow avalanche activity using detailed physics based snowpack simulations." *Geophysical Research Letters* 43.11 (2016): 5732-5740

Würzer, S, Wever, N., Juras R., Lehning, M. & Jonas, T. (2017). "Modelling liquid water transport in snow under rain-on-snow conditions–considering preferential flow." *Hydrology and Earth System Sciences* 21.3 (2017): 1741-1756

---

## Author Comment (AC2)

**Reviewer 2**

This study used Sentinel-1 SAR data to map snow wetness in alpine areas, finding strong correlations between backscatter and modeled liquid water content, as well as good agreement with wet snow avalanche occurrences (excluding the first wet snow avalanche surge). The results suggest Sentinel-1 has potential for monitoring wet snow avalanche preconditioning, particularly with increased temporal resolution (starting from additional satellite tracks). The paper is in general well written and the use of detailed avalanche catalogue to find a correlation with the Sentinel-1 backscattering is very interesting, even the study area is relatively small. There are some methodological choices that requires further explanation and discussion before the paper can be published in TC.

Dear Reviewer 2,

We sincerely appreciate your time and effort in reviewing our manuscript and providing such detailed and constructive feedback. Your comments raised important methodological and theoretical considerations that will significantly enhance the clarity and rigor of our study. Below, we respond to each point and outline the corresponding revisions we will make (colour-coding: blue review comment, black answer statement).

**Major Comments**

**1. Assessment of Radiometric Terrain Flattening and Use of Multi-Track Composites**

I expected the authors to demonstrate the added value of radiometric terrain flattening before generating the LRW mosaic, utilizing all four available Sentinel-1 tracks over the study area. Given the rapid temporal changes in snowpack LWC, as acknowledged by the authors, averaging morning and late afternoon acquisitions (as done in the LRW approach, which is essentially a weighted average) may not be optimal. For example, if a morning acquisition has a higher weight, and the LWC is low due to a cold night (resulting in higher backscatter), this could skew the result. This is particularly problematic early in the season, when the afternoon acquisitions can be affected by rapid temperature and radiation drops, showing an already potentially low LWC (and therefore backscattering) from its (midday) peak (which may be the cause of the wet snow activities in April?). To rigorously assess Sentinel-1 ability to detect the initial wet snow avalanche surge, these temporal variations should be analyzed before constructing the LRW mosaic. This analysis would provide a stronger basis for your conclusions. Therefore, the rationale behind using a mosaic with varying timestamps to address layover and shadows at a specific time requires further clarification.

We acknowledge the reviewer's concern regarding the potential biases introduced by averaging Sentinel-1 acquisitions with varying timestamps, particularly given the rapid diurnal changes in snowpack liquid water content (LWC). While, in theory, this is a critical point, our empirical experience suggests that its practical impact is less pronounced. Nevertheless, to strengthen our conclusions, in the revised version we will include an additional analysis using all four available Sentinel-1 tracks to assess the temporal evolution of $\gamma^o$ at the WFJ station before mosaicking into an LRW. This will provide insight into the effectiveness of the terrain correction and allow us to quantify potential biases introduced by merging acquisitions from different times of day.

**2. Analysis of Angular Dependencies Between Tracks**

Furthermore, analyzing the four individual tracks prior to mosaicking would provide valuable insight into the effectiveness of the terrain flattening. Residual angular dependencies, especially on aspect angle (the angle between azimuth direction of Sentinel-1 and geographic north), can introduce biases between backscatter acquired from different tracks. Has this been addressed? I recommend showing

the temporal evolution of γ° for all four tracks over the WFJ station to demonstrate the effectiveness of the terrain correction (in theory no bias should be visible between the tracks). Be aware that ascending and descending acquistions have generally a specular aspect angle.

We appreciate the suggestion to analyze angular dependencies and their potential influence on backscatter variation before mosaicking. To address this, we will include a visualization of the temporal evolution of γ° for all four tracks at the WFJ station. This will allow us to assess whether the terrain correction effectively minimizes residual angular dependencies, particularly concerning aspect angle effects. However, we expect that this test will not provide key insights on the matter, as the stations are inherently located in relatively flat terrain, where the influence of radiometric terrain correction is likely negligible.

**3. Strengthening the Literature Review**

The literature review could be strengthened by including additional background studies in both radar remote sensing of wet snow e.g. Murfitt et al., (2024) and wet snow avalanches e.g., Mitterer, and Schweizer (2013). In particular, citing key foundational works and recent publications would provide important context and allow for a more robust comparison with your results (see detail comments). This would also better support why in this study the Copernicus wet snow products were excluded a-priori.

We acknowledge that the manuscript would benefit from a more structured and focused literature review. As suggested, we will refine the discussion by:

- Removing references to dry snow backscatter changes and concentrating on wet snow literature.
- Including foundational works by Mätzler, Ulaby, and colleagues that describe backscattering mechanisms in wet snow.
- Citing more recent studies, including Murfitt et al. (2024), Picard et al. (2022), and Hendrick et al. (2024) and further recently published works.
- Investigating the connection between "first wetting" (Hendrick et al., 2024) and melting phases in SAR multitemporal data (Marin et al., 2020), as this could provide valuable theoretical insights.

- **Clarifying the Study Scope and Adjusting the Title**

While the application of a detailed avalanche catalog and Sentinel-1 backscattering time series is a novel aspect of this study, the title is misleading. The paper appears to be an exploratory investigation into the correlation between wet snow avalanches and LRW composites. Since a more robust justification for using LRW composites as the primary metric for snow wetness evolution is needed, I suggest to better sharpen the current title.

We understand that the current title may not fully reflect the study's primary focus. To better align with the manuscript's objectives, we propose adjusting the title to:
 **"Extracting Wet Snow Avalanche Precondition Information from Sentinel-1 Multi-Track Composites"**
 This revised title more accurately conveys that our study is exploratory and focuses on Sentinel-1's potential for monitoring avalanche preconditioning rather than providing a validated operational product.

**5. Ensuring Compliance with Open Science Standards**

Consistent with TC publications policy, I suggest the authors to make the LRW time series and all data publicly accessible. The statement "data available upon request" does not meet current standards for open science and reproducibility. Depositing the data in a recognized public repository (e.g.,

ENVIDAT or Zenodo, or Dryad) would greatly enhance the value and impact of this work by enabling independent verification and reuse of the data.

We fully support open data policies and recognize the importance of making our datasets publicly available. As noted in our response to Reviewer 1, we will:

- Deposit our code in GitHub.
- Archive all data in EnviDat, ensuring compliance with *The Cryosphere*'s data policy.

**Detailed Comments & Minor Revisions**

L17: more recent works have been published on how use Sentinel-1 for SWE/runoff modeling e.g., Cluzet et al. (2024) or Premier et al. (2023).

Line 17          We will update the literature references to include more recent studies, such as Cluzet et al. (2024) and Premier et al. (2023), on using Sentinel-1 for SWE/runoff modeling.

L23 to 34: This section would benefit from a more focused literature review. The current "ping-pong" between dry and wet snow literature makes it difficult to follow the narrative. Since the paper focus is on wet snow, I recommend removing the discussion of dry snow backscatter changes and concentrating on a comprehensive review of wet snow literature. Key works by Matzler, Ulaby and colleagues that describe the main backscattering mechanisms in wet snow should be included, as well as recent advancements e.g., Picard et al. (2022) or Murfitt et al. (2024). I also suggest exploring the potential link between the "first wetting" described by Hendrick et al. (2024) and the melting phases presented in Marin et al. (2020) for SAR multitemporal data. Investigating this connection could offer valuable theoretical insights.

Line 23-34       The literature review will be revised to:

- Remove discussions of dry snow backscatter changes.
- Structure the review to focus on wet snow SAR applications.
- Cite additional key references as suggested.
- L41: Two angles affect backscatter: local incidence angle and aspect angle. It is crucial to consider both, as they have distinct effects. To ensure the terrain flattening effectively corrects for these influences, please clarify whether the aspect angle was incorporated into the process. Showing the gamma naught ($\gamma^o$) values for all four tracks would be very helpful in identifying any residual biases between them.

Line 41          We will clarify the distinction between local incidence angle and aspect angle in terrain flattening. To visually assess residual biases, we will include an additional figure showing the mean temporal $\gamma^o$ values for all four Sentinel-1 tracks at the station WFJ.

L43-45: While I understand the intent of averaging to minimize noise, I believe it is important to consider that the identified "outliers" could represent real-world afternoon wet snow conditions not captured in the morning data. This raises concerns about potentially losing valuable temporal information. I also disagree with the assertion that multi-temporal averaging improves temporal resolution; by combining data from different times, it effectively lowers the resolution. Perhaps exploring alternative noise reduction techniques that preserve temporal fidelity would be beneficial.

Line 43-45       We agree that multi-temporal averaging does not improve temporal resolution but rather increases the effective revisit frequency by utilizing all four available tracks.

The added value of higher revisit frequency is only given if we use more than one ascending or descending track, we will clarify this point and emphasize that our approach enhances data availability rather than true temporal resolution.

L50: The use of a 5x5 meter resolution raises concerns, as the original Sentinel-1 data has a ~5x15 meter resolution. This upsampling introduces artificial detail and does not represent true information gain. A 20x20 meter resolution would be more consistent with the original data. Could you please justify the decision to use 5x5 meters and explain how this upsampling was handled?

Line 50       The decision to use a 5×5 m resolution will be explicitly justified. We will clarify that:

- The original Sentinel-1 resolution is ~5×15 m, meaning the upsampling does as stated by the author does not add information but simply aligns with a higher-resolution DEM thereby becoming closer to point measurements – which is our reference.
- We performed local validation using both a 3×3 window and single-pixel validation at station locations to ensure that upsampling did not introduce artifacts.

Line 69: please use the symbol $\gamma^o$ or gamma nought instead of gamma0

Line 69       We will ensure that $\gamma^o$ (gamma naught) is consistently used instead of "gamma0" throughout the manuscript and enhance to differentiate between $\gamma^o_T$ (Radiometric terrain (RTC) corrected backscatter) and $\gamma^o_{LRW}$ (Weighted sum of contributing RTC backscatter value)

Line 70: To ensure the validity of the results, please provide a more detailed description of how the gamma software processes $\gamma^o$. Has the output of the gamma software been compared and validated against the data elaborated by David Small? The original Small et al. (2022) paper highlights potential inaccuracies in SNAP (an ESA software) due to different implementations (see the end of Section II-B). This raises concerns about potential similar discrepancies. A direct comparison or a comment on this would be beneficial for the community.

Line 70       We appreciate the suggestion to provide further details on how $\gamma^o$ processing in Gamma software was conducted. The implementation of the radiometric correction into follows Small et al., 2022. As stated in the acknowledgements, we also worked in close collaboration with David Small and our products have been initially validated against products generated by Small et al. (2022).

L126: it is not clear how you perform the average and why.

Line 126       We will provide a clearer explanation of how the averaging process was conducted in SNOWPACK. Specifically, we will state that:
The mean was calculated using the two SNOWPACK values closest to the ascending and descending acquisition times, ensuring consistency with Sentinel-1 observations.

Results section: A comparison of the proposed processing with established methods, such as those of Mitterer and Schweizer (2013), Bellaire et al. (2017), and Hendrick et al. (2024), would greatly enhance the manuscript by demonstrating the novelty and performance of the proposed approach.

Results Section: We acknowledge the suggestion to compare our approach with Mitterer and Schweizer (2013), Bellaire et al. (2017), and Hendrick et al. (2024).

However, this analysis aims to demonstrate the potential for integrating the system into models like SNOWPACK in the future, once the increasing availability of freely accessible SAR data provides sufficiently high temporal resolution. Currently, and as discussed in the manuscript the temporal resolution remains too low to integrate a

remote sensing product spanning several days into a model with 3-hourly predictions. Given that our study is exploratory, we do not directly integrate our approach into existing operational workflows. So instead, we will emphasize that our method provides a potential future complement to model-based LWC estimates (such as SNOWPACK), which are spatially restricted (e.g., Swiss nationwide applications) or less resolved than the models available in such high temporal resolution.

Figure 5: The gray dots are difficult to see and could benefit from increased contrast or a different color.

Figure 5         We will improve readability by adjusting the colour of the dots to enhance visibility.

Additional references:

Sascha Bellaire, Alec van Herwijnen, Christoph Mitterer, Jürg Schweizer, On forecasting wet-snow avalanche activity using simulated snow cover data, Cold Regions Science and Technology, Volume 144, 2017, https://doi.org/10.1016/j.coldregions.2017.09.013.

Cluzet, B., Magnusson, J., Quéno, L., Mazzotti, G., Mott, R., and Jonas, T.: Exploring how Sentinel-1 wet-snow maps can inform fully distributed physically based snowpack models, The Cryosphere, 18, 5753–5767, https://doi.org/10.5194/tc-18-5753-2024, 2024.

Marin, C., Bertoldi, G., Premier, V., Callegari, M., Brida, C., Hürkamp, K., Tschiersch, J., Zebisch, M., and Notarnicola, C.: Use of Sentinel-1 radar observations to evaluate snowmelt dynamics in alpine regions, The Cryosphere, 14, 935–956, https://doi.org/10.5194/tc-14-935-2020, 2020.

Mitterer, C. and Schweizer, J.: Analysis of the snow-atmosphere energy balance during wet-snow instabilities and implications for avalanche prediction, The Cryosphere, 7, 205–216, https://doi.org/10.5194/tc-7-205-2013, 2013.

Murfitt, J., Duguay, C., Picard, G., and Lemmetyinen, J.: Forward modelling of synthetic-aperture radar (SAR) backscatter during lake ice melt conditions using the Snow Microwave Radiative Transfer (SMRT) model, The Cryosphere, 18, 869–888, https://doi.org/10.5194/tc-18-869-2024, 2024.

Picard, G., Leduc-Leballeur, M., Banwell, A. F., Brucker, L., and Macelloni, G.: The sensitivity of satellite microwave observations to liquid water in the Antarctic snowpack, The Cryosphere, 16, 5061–5083, https://doi.org/10.5194/tc-16-5061-2022, 2022.

Premier, V., Marin, C., Bertoldi, G., Barella, R., Notarnicola, C., and Bruzzone, L.: Exploring the use of multi-source high-resolution satellite data for snow water equivalent reconstruction over mountainous catchments, The Cryosphere, 17, 2387–2407, https://doi.org/10.5194/tc-17-2387-2023, 2023.

**Citation**: https://doi.org/10.5194/egusphere-2024-1510-RC2

References         We will make sure to implement the suggested literature where appropriate and generally strengthen our literature review as mentioned earlier